# Decoding Nucleotide Repeat Expansion Diseases: Novel Insights from *Drosophila melanogaster* Studies

**DOI:** 10.3390/ijms252111794

**Published:** 2024-11-02

**Authors:** Sandra Atienzar-Aroca, Marleen Kat, Arturo López-Castel

**Affiliations:** 1Department of Dentristy, Faculty of Health Sciences, European University of Valencia, 46010 Valencia, Spain; sandra.atienza@universidadeuropea.es; 2Institute for Life Sciences and Chemistry, HU University of Applied Sciences Utrecht, NL-3584 Utrecht, The Netherlands; marleen.kat@gmail.com; 3Human Translational Genomics Group, University Research Institute for Biotechnology and Biomedicine (BIOTECMED), Universidad de Valencia, 46100 Burjasot, Spain; 4INCLIVA Biomedical Research Institute, 46010 Valencia, Spain; 5CIBERER, Centro de Investigación en Red de Enfermedades Raras, Instituto de Salud Carlos III, 28029 Madrid, Spain

**Keywords:** repeat expansion, rare disease, *Drosophila melanogaster*, high-throughput screening, protein aggregation, toxic RNA, protein loss-of-function

## Abstract

*Drosophila melanogaster* usage has provided substantial insights into the pathogenesis of several nucleotide repeat expansion diseases (NREDs), a group of genetic diseases characterized by the abnormal expansion of DNA repeats. Leveraging the genetic simplicity and manipulability of Drosophila, researchers have successfully modeled close to 15 NREDs such as Huntington’s disease (HD), several spinocerebellar ataxias (SCA), and myotonic dystrophies type 1 and 2 (DM1/DM2). These models have been instrumental in characterizing the principal associated molecular mechanisms: protein aggregation, RNA toxicity, and protein function loss, thus recapitulating key features of human disease. Used in chemical and genetic screenings, they also enable us to identify promising small molecules and genetic modifiers that mitigate the toxic effects of expanded repeats. This review summarizes the close to 150 studies performed in this area during the last seven years. The relevant highlights are the achievement of the first fly-based models for some NREDs, the incorporation of new technologies such as CRISPR for developing or evaluating transgenic flies containing repeat expanded motifs, and the evaluation of less understood toxic mechanisms in NREDs such as RAN translation. Overall, *Drosophila melanogaster* remains a powerful platform for research in NREDs.

## 1. Introduction

The human genome harbors over one million annotated tandem repetitive (or repeat) sequences (TRs), classified into distinct categories depending on their genomic localization, frequency of occurrence, and length [1]. Among the TR types with the highest presence throughout the human genome, microsatellites display short repetitive motifs (with definitions varying slightly between 1–6 or up to 10 base pairs) and are thus also known as short tandem repeats (STRs). Notably, database analyses indicate a prevalence of A/T repeats among mononucleotides and CA/GT repeats among dinucleotide repetitions, while GC-rich repeats are disproportionately abundant in trinucleotide and tetranucleotide expansion motifs [2]. STRs are typically repeated 5–50 times and are commonly located in non-coding, non-genic regions, able to accumulate unhindered changes in the repeat size over generations that provide well-characterized, neutral variability to the human genome and are used for forensic studies in populations [3]. However, when the STR is located in a gene (in either the regulatory flanking region, intron, or exon) the repeat size can switch to uncontrolled expansion, directly leading to a growing list of diseases, grouped under the nucleotide repeat expansion disorder (NRED) abbreviation [4,5].

The early 1990’s saw the discovery of the first STRs connected to human pathology, commonly involving trinucleotide motifs (trinucleotide repeat expansion diseases, TREDs) present in both coding and non-coding regions of the genome. The first genetic expansions (identified in 1991) consisted of CGG repeats in the 5′untranslated region of the fragile X messenger ribonucleoprotein 1 (*FMR1*) gene, associated with fragile X syndrome (FXS), and CAG repeats in the first exon of the androgen receptor (*AR*) gene, linked to spinal and bulbar muscular atrophy (SBMA). Since then, expansions of CAG, CTG, GAA, CCG, and CGG trinucleotide repeat motifs, but also sequences from di-, tetra-, penta-, and hexanucleotides, have been linked to the over 70 currently identified NREDs [reviewed in 5]. NREDs are mostly linked to rare diseases, often connected to neuromuscular and neurodegeneration symptoms, such as myotonic dystrophies (DM1 and DM2), Huntington’s disease (HD), or distinct types of spinocerebellar ataxias (SCAs), but some are also associated with intellectual disability, such as fragile X syndrome (FX) or Friedreich’s ataxia (FA) [6]. Several NREDs have been well-characterized for the last three decades, allowing us to distinguish three main pathways for the cellular toxicity and dysregulation linked to the heterogeneous clinical profiles finally observed in patients [7] (Figure 1). The first pathway involves the loss-of-function of mutated genes due to silenced or reduced gene expression, as observed in disorders like FXS and FRDA. A second one, caused by the presence of RNA gain-of-function or toxic RNA (RNAtox), has been suggested in disorders like DM1/DM2 and spinocerebellar ataxias types 8, 10, 31, and 36, from the transcribed but not translated expanded RNA repeats from the mutated gene. These RNAs containing expanded repeats (RNAexp) have been shown to accumulate as RNA foci in different cell types and dysregulate important RNA-binding proteins (RBPs), resulting in their loss of function. As a subclass of the RNAtox mechanism, it is recognized that toxicity comes from the transcripts from sense strand, although a role for the anti-sense transcripts is still under debate, as their presence has been detected across repeat expansion in several NREDs [8]. The third main pathway is the polyprotein gain-of-function from regular translation from repeat expanded alleles but also suggested from unconventional repeat-associated non-AUG translation (RAN translation) [9]. Relevant to the study of NREDs is their frequent, simultaneous display of previously defined mechanisms. This is exemplified by ALS/FTD caused by a repeat expansion in the *c9orf72* gene region, displaying huge levels of complexity from the molecular to the clinical level.

The discovery of the repeat expansion mutation type was reached via different approaches, initially involving older technologies such as positional cloning, cytogenetic mapping, Sanger sequencing, or Southern blot (reviewed in [5]). However, more complex repeat sequences, for example, because of GC-rich repeat-motif sequences and forming stable DNA structures, are very difficult to amplify by regular PCR application. Only recently, new generation sequencing (NGS) advances have enabled us to report new disease-causing TR expansions, due to the ability to generate unbiased long sequencing runs of DNA without fragmentation from novel Nanopore or PacBio technology, making them an invaluable tool in genomic research [5,10].

*Drosophila melanogaster* is a small animal model that is outstanding for use in disease-based studies due to its simpler, less redundant genetics compared to mammal species and the availability of an enormous and customizable number of transgenic mutants, impractical in mammals [11]. Its genetic flexibility, coupled with resources such as FlyBase and the Bloomington Drosophila Stock Center [12,13], has revolutionized genome editing in *Drosophila*, enabling the precise study of the function of specific genes in different disease areas. Consequently, fruit flies have also proven to be an invaluable model for studying the underpinning of human diseases at the laboratory level [14], including NREDs [15,16]. Nonetheless, the generation of new data around NREDs is rapidly growing, as are the tools linked to this research. Therefore, an update on how the *Drosophila* field is evolving in terms of NRED models is highly relevant. In the following sections, we review aspects covered using previously published NRED fly models, also identifying new NREDs modeled in flies, as well as outlining how these models are involved in therapeutic advancement.

## 2. Understanding NREDs Using *Drosophila melanogaster*: A 2018–2024 Update

### 2.1. Studies on Previously Targeted NREDs

#### 2.1.1. Use of Fly-Based Models in NRDEs with Gain-of-Function Protein Mechanism 

A significant number of neurodegenerative diseases share abnormal aggregation of polyglutamine (polyQ) protein due to a shared CAG repeat instability in the coding region of a specific gene [17]. This toxic gain-of-function (GoF) protein phenomenon occurs when the repeat exceeds a pathogenic threshold of >35 CAG units, triggering abnormal misfold of the resultant polyQ protein, causing aggregation and thus disrupting cellular functions, mainly in specific neuronal subpopulations [18] (Figure 1). Previous *Drosophila* models have already been developed for polyQ NREDs, like Huntington’s disease (HD), several spinocerebellar ataxias (SCAs, types 1, 3, 6, 7, and 17), spinobulbar muscular atrophy (SBMA), and dentatorubal-pallidoluysian atrophy (DRPLA) [reviewed in 15-16]. However, what has been explored in flies for polyQ diseases since then?

The ability to detect a disease at a presymptomatic stage is an exciting area of research. The use of **HD** flies expressing a 93-repeat polyglutamine expansion (HttQ93) in neurons uncovered unique metabolites when neuropathologic phenotypes were present compared with non-HD flies, thus achieving good discrimination between diseased flies and controls. More interestingly, a distinct list of altered metabolites was observed between pre-symptomatic (10-day-old) and symptomatic (16-day-old) HD flies. These results point to metabolism perturbations changing dramatically during disease development [19]. The reproducibility of similar results in other HD models and humans could open a line of research to define biomarkers to follow the progression of HD brain pathology. A different study using the same HttQ93 flies explored cellular/genetic mechanisms connected to HD pathology, suggesting a potentially important role for autophagy. It was documented that modulating the expression level of the *gs1* gene (overexpression) was able to promote autophagy and improved HD-linked phenotypes [20]. This was achieved by triggering changes in amino acid levels and inducing a starvation-like condition. This mechanism, well-characterized in flies, is crucial for eliminating toxic aggregates through preventing TOR activation and phosphorylation of S6K [20]. A similar fly HD model, but generating polyglutamine aggregation from a 128 polyQ expansion, also targeted a different autophagy-linked gene, *atg8*. Again, overexpression of an autophagy gene was correlated with the mitigation of HD side effects, such as circadian rhythm and sleep disturbances observed in the flies, although polyprotein aggregate formation was not prevented [21]. Together, all these results suggest an important transcriptional dysregulation in HD, which was confirmed via transcriptional study, at mRNA and microRNA levels, with HD flies. The results showed that 32 miRNAs significantly changed levels in head samples of HD flies [22]. The comparison of miRNA and mRNA expression data detected correlations in the impacted molecular pathways by characterizing putative targets of almost all dysregulated miRNAs overrepresented among the upregulated mRNAs. A further functional analysis connected the transcriptional alterations to key lipolytic and lipogenic genes [22]. Interestingly, a different approach connects with these latter results by showing that fly fat body undergoes extensive alteration and eventually surrenders to increased apoptotic cell death in terminal stages of the disease [22]. Otherwise, although HD is widely recognized as a disease affecting the nervous system, early evidence suggests that peripheral and non-neuronal tissues are also affected. Comparing Q72 (pathogenic) or Q25 (non-pathogenic) expression in fly muscles, researchers distinguished detrimental phenotypes such as reduced lifespan, decreased locomotion, and accumulation of protein aggregates in this tissue for the largest polyQ [23] (Table 1). Strikingly, in HD fly expression of *Hsp70*, a well-documented suppressor of polyglutamine aggregates, greatly reduced the accumulation of aggregates in the eye but did not prevent reduction of lifespan when expressed in the muscle. These results suggest that molecular mechanisms underlying the detrimental effects of aggregates in the muscle could be different from the ones involved in the nervous system [23]. All these findings underscore the complexity of HD and the importance of understanding not only the mechanisms of toxicity associated with polyQ expansions but also the secondary cellular pathways that can influence disease progression. Additional protective genes found or validated in HD flies are *rab8* and *yod1* genes, which, when overexpressed, possess the ability to modulate the number or size of toxic aggregates [24,25]. Taking advantage of the new functional protocols established with flies, researchers used a high-resolution respirometry approach able to evaluate different mitochondrial processes, such as oxidative phosphorylation and electron transfer capacity, to uncover dysregulation of the processes when a mutant *Htt* gene fragment targeted neurons or muscles. Interestingly, these defects can be ameliorated by enhancing mitochondrial function, for example, co-expression of parkin, an E3 ubiquitin ligase critical for mitochondrial dynamics, resulting in significant rescue of neurodegeneration, viability, and longevity in HD model flies [26]. An original research question being explored concerns polyprotein structure determination. PolyQ sequences up to Q36 in *Htt* gene are not known to be toxic, while lengths above this almost invariably lead to increased disease risk and decreased age of onset. Recently, many physical states (monomers, dimers, tetramers, non-β oligomers, nanofibrils, and clustered amyloid fibrils) have been characterized on the self-association landscape of polyQ, making it difficult to identify the true molecular species responsible for HD toxicity [27]. To gain information in this area, various *htt-ex1* sequences (with and without the addition of repeat breaker mutations) were expressed in *Drosophila* models of HD. In both cases, strong aggregation was observed despite the presence of short polyQ repeat lengths, and this proved toxic, dramatically breaking the repeat length paradigm, suggesting that aggregates with the fundamental amyloid folding motif are likely the main toxic species in HD [28] (Table 1). Another key aspect proposed in HD is connected to the biologic function of the HTT protein. An important role for vesicular transport within axons has been suggested, but nothing is known about the cargo that HTT transports to/from synapses. The development of a fly-based dual-color imaging sensor revealed that HTT and Rab4 move together on a unique putative vesicle that may also contain synaptotagmin, synaptobrevin, and Rab11 factors. Pathogenic (expanded) HTT disrupts the motility of HTT-Rab4 and results in larval locomotion defects, aberrant synaptic morphology, and decreased lifespan of HD flies, which are rescued by an excess of Rab4 [29]. Consistent with these observations, RAB4 motility is perturbed in neurons differentiated from iPSCs derived from human HD patients [29]. As a novelty, the epigenetic modulation of HD pathology has been proposed as a new mechanism to be studied [22]. Comparing flies expressing HttQ120 or HttQ25 has revealed altered expression of microRNAs such as mir-10, mir-137, and mir-1010 and identified post-translational histone mutations such as H3.3K14 [22,30]. Although HD development is known to be connected to the presence of an expanded polyQ repeat, the exact mechanism is still not fully understood. Recently, by **creating different new HD flies** containing sequences of the HTT human gene (full, only exon 1, or with small changes in the sequence around the repeat), it has been observed that polyQ expansion is necessary to promote the formation of toxic aggregates (Table 1). However, this alone is not sufficient to induce pathology, demonstrating the key role of genomic context in disease manifestation [31]. Finally, a recent fly-based study identified the junctophilin gene (*jp*) as a modifier of neuronal degeneration [32], taking advantage of the genomic simplicity of flies, which possess only one version of the *jp* gene instead of the four observed in humans, and uncovering an unsuspected ancestral functional relationship with the Notch pathway.

Dominant autosomal spinocerebellar ataxias (**SCAs**) constitute a diverse group of neurodegenerative disorders characterized by progressive loss of motor coordination and balance, several also caused by the expansion of CAG nucleotide repeats in the coding region of specific genes, spawning toxic polyQs [33,34] (Figure 1). Despite their common genetic cause, each SCA type has a different gene location of the repeat and presents unique clinical features, potentially differing in certain mechanisms involved in pathology progression. In **SCA1**, a cross-species screen (human cells vs. *Drosophila*) identified 22 regulators of the mutant ATXN1 protein aggregation. One of these, the transglutaminase 5 (TG5), preferentially regulated mutant ATXN1 over the wild-type (wt) protein by strong modulation of protein stability and oligomerization. This finding was validated in flies, where perturbing the *tg* (*TG5* homolog) endogenous gene modulated mutant ATXN1 toxicity [35]. Similarly, the PAK1 kinase pathway has also been revealed as important in SCA1. Mutations inducing a functional reduction of PAK1 and PAK3 proteins have been found to modulate ATXN1 levels and, in flies, to reduce neurotoxicity by improving phenotypes of retinal degeneration and motor performance [36]. The identification of new genetic players in SCA1 toxicity mechanisms has recently progressed using data sets previously generated in disease models including *Drosophila*. One interesting source is EvoPPI3, a meta-database for protein–protein interactions (PPI), which integrates data for nine neurodegenerative polyQ diseases. The available fly-based datasets for wt and expanded *ataxin-1* mutants showed that the ataxin-1 human network is much larger than previously thought, moving from 380 to 909 interactors, of which 16 are putative novel SCA1 therapeutic targets, mainly involved in binding and catalytic activities [37]. A functional study in an ATXN1-Q82 *Drosophila* model found reduced levels of the ATM homolog gene, linked to ameliorated motor symptoms [38], thus validating the involvement of DNA damage repair in modifying the age of disease symptoms onset in SCA1 and suggesting a similar result in HD [39]. These similarities between the *ataxin-1* and the *huntingtin* responses to DNA damage provide further support for a shared pathogenic mechanism for polyglutamine expansion diseases. Examining **SCA3**, the fly disease models were recently useful at different levels. From a cell-based screen of >2,000 siRNAs targeting druggable human genes, 15 of the hits found were validated in a SCA3 *Drosophila* model, characterizing orthologs of *CHD4, FBXL3*, *HR,* and *MC3R* able to regulate mutant ATXN3-mediated toxicity in fly eyes [40]. Similar fly-based approaches explored the search for additional modulators of the insoluble protein aggregates in SCA3 expressing neurons. For example, a genetic screen performed in 19 conserved genes from the major components of the Cullin-RING ubiquitin ligases (CRLs) family, specifically implicated in polyQ pathogenesis, taking advantage of the high evolutionary conservation of this family in the *Drosophila* genome, identified the F-box gene (FipoQ) as involved in polyQ pathogenesis [41]. Similarly, an eye-specific siRNA expression screening in a SCA3-polyQ78 fly model identified genes that can enhance or suppress eye degeneration. Selective expression inhibition of the *relish* gene (a conserved NF-κB transcription factor), specifically in astrocytes, extended the lifespan of diseased flies [42]. This type of in vivo approach and the findings contribute to a better understanding of SCA3 disease mechanisms and provide insights for developing promising treatments. A different approach with SCA3 flies has explored the roles of the ATXN3 protein. Researchers compared the toxicity of two major isoforms found for the ataxin-3 factor (Table 1). The study revealed that isoform 1, which predominates in the human brain, was notably more toxic in *Drosophila* due to its lower degradation rate and higher tendency to form aggregates. Conversely, isoform 2 degrades more rapidly through the proteasome, resulting in lower protein levels and less aggregation, which diminishes its toxicity [43]. These results link a better understanding of the role of the different domains of Atxn3 with the pathogenesis of the ubiquitin-binding site 1 (UbS1) in its catalytic domain, which is suggested to be important. Transgenic *Drosophila melanogaster* lines, which express polyQ expanded Atxn3 without a functional UbS1, markedly exacerbate in vivo toxicity, not by affecting its aggregation or sub-cellular localization but rather by impacting on its ubiquitin active role [43]. It should be mentioned that fly models are not perfect and disease recapitulations are not always possible, or still not achieved, as was concluded from a study testing *Drosophila* SCA3 models in which RAN translation was not detected [44]. However, this small animal can easily provide readings of complex functional features. SCA3 is associated with distinct, altered gait and tremor movements reflective of the underlying disease etiology. In a recent fly-based study, the researchers developed a machine-learning image-analysis program called FLLIT, which was able to automatically track leg claw positions of freely moving flies and record them on high-speed video. The result is a series of gait measurements, markedly different in fly SCA3 models, which exhibited walking gait and tremor signatures, recapitulating characteristics of the respective human disease [45]. This novel in vivo approach opens up its regular use to dissect the neurogenetic mechanisms underlying movement disorders in several pathologies. In SCA7, new *Drosophila* models have been developed and characterized. They feature ectopic expression of the full-length human ataxin-7 (ATXN7) protein, tagged with *Myc* and containing varying lengths of polyglutamines, including normal (Q10) and expanded (Q92). Both stocks exhibit variable and tissue-specific toxicity, but the Q92 protein shows progressively stronger effects on retinal cell stability and longevity [46] (Table 1). Regarding SCA17, new Drosophila transgenic lines have also been introduced in recent years. Transgenic flies were developed that express the human, full-length TATA box binding protein (TBP) with CAG/CAA repeats encoding either Q25 (wild-type) or Q63 (mutant) on a uniform genetic background. Upon examination of their expression, it is observed that developmental issues are induced, and survival and mobility in adult flies are decreased. The mutant Q63 variant exhibits higher toxicity and propensity for protein aggregation compared to the wild-type [47] (Table 1). Additionally, the interaction of hTBP expanded protein with other homeoproteins was performed in the fly brain using targeted expression with the system UAS/GAL4. The results displayed interaction through its glutamine-rich region, and hTBP protein aggregates were found to affect the locomotor capacity of flies [48]. All these new models currently offer a highly flexible and controlled approach for investigating the underlying mechanisms of SCAs and the possibility of discovering therapeutic opportunities.

Spinal and bulbar muscular atrophy (SBMA) stems from an aberrant expansion within the polyglutamine repeat sequence of the androgen receptor (*AR*) gene [17]. To better conform with established mammalian models and human pathology, new Drosophila models of SBMA were established [49] (Table 1). These fresh models express the full-length, human *AR* gene in both the wild-type (*wt*) configuration and the pathologically expanded-CAG repeat configuration. Contrasting with the previous models, the new ones were constructed using pHiC31-dependent integration to be able to insert both the WT and expanded AR transgenes into a well-known secure locus. This approach will facilitate direct comparisons among the new lines and any others generated in a similar fashion, mitigating potential off-target effects of random integration or variations in expression stemming from differences in transgene copy numbers.

Oculopharyngeal muscular dystrophy (OPMD) is the only one of the eight NRED human disorders linked to the presence of a toxic polyalanine (polyA), resulting from expansion mutations in GCN tracts [50] and previously modeled in flies [51]. A more recent study using these flies performed a genome-wide and targeted genetic screen, with results suggesting a role for the UPS pathway in OPMD. Mechanistically, lowered proteasome activity is linked with reduced OPMD muscle defects, suggesting a pathological increase in UPS activity in the disease [52]. Notably, no other studies or new models of polyA have been identified using *Drosophila melanogaster*.

**Table 1 ijms-25-11794-t001:** Fly-based models for NRED diseases referenced as new-fangled during the period of time reviewed.

Disease Pathway	Fly Model	Disease	Repeat Exp.	Main Features	Ref.
**Protein** **GoF**	Httex1Q72	HD	CAG	*Drosophila* UAS lines each containing a transgene with an N-terminal fragment of human *huntingtin* with 72 glutamine (Q72) repeats crossed with DJ694, MHC, or Mef2-Gal4 stocks driving repeat expression to the muscle.	[23]
	Httex1-βHP Httex1-βHP-P	HD	CAG	Httex1 fly UAS models involving β-turn promoting motifs, as well as β-turn motifs with inclusion of β-breaker residues. Moreover, double crossing with Httex1-Q25, 46, and 97 were evaluated.	[28]
	Fl-Htt Httex1 Htt118 Htt117	HD	CAG	*Drosophila* UAS lines including fl-HTT as well as HTTex1 (90 amino acids), HTT118 (N-terminal human huntingtin fragment with an expanded polyglutamine repeat under the control of the endogenous human promoter (shortstop)), and HTT171 (N-terminal fragment with 171 amino acids) in all cases with either 120 or 25 Qs	[31]
	Atxn3-isf1 Atxn3-isf2	SCA3	CAG	Fly UAS lines containing human ataxin-3 cDNA (isoform 1 or 2) with a CAGCAA repeat that encodes 80Q. With a CAGCAA repeat instead of a pure CAG tract to circumvent the possibility of RAN translation and mRNA toxicity	[43]
	Atxn7-Q92	SCA7	CAG	*Drosophila* transgenic lines with full-length, Myc-tagged, human ataxin-7 (ATXN7) protein with normal (Q10) or expanded (Q92) polyQ from a pure CAG	[46]
	Atxn17	SCA17	CAG	*Drosophila* transgenic lines of SCA17 that express HA-tagged, full-length human *TBP* with CAG/CAA repeats encoding either Q25 (wild-type) or Q63 (SCA17) with the UAS-Gal4 system	[47]
	AR(SBMA)	SBMA	CAG	Drosophila lines including full-length human *AR* with either wild-type (Q20) or expanded (Q112) repeats from a mixed CAG/CAA repeat sequence	[49]
	DM1–250CTG	DM1	CTG	*Drosophila* DM1 lines bearing UAS-iCTG constructs with 240, 480, 600, and 960 CTGs crossed with Hand-Gal4 (cardiac drivers)	[53]
**RNA tox**	DM1–250CTG	DM1	CTG	Fly line with pure UAS-250 CTG	[53]
	DM1–1100CCTG	DM2	CCTG	Fly line with pure UAS-1100 CCTG	[53]
	DM2–106	DM2	CCTG	Fly line with pure UAS-106CCTG	[54]
RNA tox	RNAi-Jp * overexp-Jp *	HDL2	CTG	Flies were induced with up- (RNAi-VIE-260B, whose target sequence is contained in an exon that is common to the four major isoforms of the jp gene) and downregulation (stock with an insertion of the P{XP} transgene within the jp locus, P{XP}jp^d04563^) of endogenous Jp in a tissue-specific manner	[32]
**Protein LoF**	fh-200GAA	FRDA	GAA	* Drosophila * transgenic line generated with CRISPR/Cas9 technology to insert a GAA expansion of approximately 200 units in the endogenous frataxin (*fh*) gene in a similar position to that in the human gene, in the first intron of the *fh* gene, and flanked by sequences from the human intron of FXN	[55]
**Protein GoF**/RNA tox	UAS-DPR1000	c9orf72 ALS/FTD	G4C2	* Drosophila * trasngenic line containing 1000 G4C2 repeats	[56]
**Protein GoF**/RNA tox	uN2CpolyG	NIID	CGG	UAS transgenic *Drosophila* model including 100 CGGs able to be translated to uN2CpolyG in multiple systems with the UAS-Gal4 system	[57]
Protein GoF/RNA tox	RNAi-D12 *	FAME4	TTTTA/TTTCA	Flies with RNAi (knockdown with Vienna GD29954 and KK105606 stocks) of *Drosophila* D12 gene (homolog of human *YEATS2*)	[58]

For the “Disease pathway” row, indicated **in bold,** the mechanism characterized in the fly model is published and here referenced. *** NRED model based on the design of flies without the manifestation of a repeat expansion motif.

#### 2.1.2. Use of Fly-Based Models in NREDs with RNA Gain-of-Function Mechanism

A key mechanism of action (MoA) in several NREDs involves toxic RNA generation from the transcription of intergenic non-coding DNA repeat expansions (Figure 1). The mutated, expanded, or toxic messenger RNA (RNAexp or RNAtox) is abnormally retained in the nucleus and detected as aggregated in punctuated ribonuclear foci. It is accepted that these foci have the ability to trigger protein dysfunction of different RNA binding proteins, either by direct interaction or by still unknown mechanisms not requiring physical RNA–protein interaction [59]. This leads to a cascade of cell pathway impairment, commonly hallmarked by the dysregulation of alternatively splicing (AS) of several genes [59]. The first disorder associated with the formation of toxic RNA was myotonic dystrophy type 1 (DM1) [60], a genetic disease linked to abnormal expansion (>50 units) of a CTG motif disease. This same repeat expansion is linked to other disorders such as new spinocerebellar ataxia, (SCA8), Huntington disease-like 2 (HDL2), and Fuchs’ endothelial corneal dystrophy (FECD) [61,62,63]. Distinct repeat motifs and/or lengths have been associated with the same MoA in other pathologies, like CGG in fragile-X-associated tremor/ataxia syndrome (FXTAS), CCTG in myotonic dystrophy type 2 (DM2), TGGAA in SCA31, and recently, a complex repeat motif, GGGGCC, (G4C2) hexanucleotide repeat in the chromosome 9 ORF 72 *(c9orf72*) gene as the most common genetic cause of amyotrophic lateral sclerosis (ALS) with frontotemporal dementia (C9-ALS/FTD) [64,65,66,67]. What is new regarding the use of fly-based models for these types of NREDs?

In some diseases like SCA8, there is an absence of studies, but more generally, we found a significant number of studies in different NREDs. Regarding DM1, “old” models, mimicking key disease dysregulations, have brought to light novel disease mechanisms. A relevant recent work indicated that restoring miR-7 levels in patient-derived cells is a candidate therapeutic target for counteracting muscle dysfunction in DM1 [68]. This study followed on from a first report from the same group showing that miR-7 was downregulated in a DM1 *Drosophila* model, a feature conserved in patient biopsies [69]. Interestingly, from the same *Drosophila* study, miR-1, a muscle- and heart-specific miRNA, was also identified as dysregulated in the DM1 flies containing 480 interrupted expanded CTGs (i(CTG)480) and was also dysregulated in DM1 muscle patients [70]. In a more recent study exploring cardiac-related features, targeted miR-1 downregulation in the heart of flies displayed dilated cardiomyopathy (DCM), a DM1-associated phenotype. Interestingly, DCM was not observed from the direct overexpression of CTG960 in flies, revealing only a slight cardiac dilation phenotype in flies [71]. From these studies, and combining fly-based in silico screening and transcriptional profiling of DM1 cardiac cells, the multiplexin (Mp) factor was identified as a new cardiac miR-1 target involved in DM1. *Mp* and its human ortholog *Col15A1* are both highly enriched in cardiac cells of DCM-developing DM1 flies and in heart samples from DM1 patients with DCM, respectively, and its attenuation rescues the DCM phenotype of aged DM1 flies [71]. In a different study, the misbalance between MBNL1 and CELF1, the two most important proteins dysregulated in DM1, was simulated in the *Drosophila* heart and an RNAseq of the cardiac cells was performed. The approach detected a few dysregulated genes, a few controlling cellular calcium levels through targeting the expression of straightjacket/*α2δ3* in the fly heart, a regulatory subunit of a voltage-gated calcium channel that led to asynchronous heartbeat, a hallmark of abnormal conduction [72]. The validation results involved straightjacket knockdown, improving these symptoms in DM1 fly models. From these results, it was also observed that ventricular *α2δ3* expression is low in healthy mice and humans, but significantly elevated in ventricular muscles from DM1 patients with conduction defects. Together, these findings suggest that reducing ventricular straightjacket/α2δ3 levels could offer a strategy to prevent conduction defects in DM1. The same research group provided solid fly-based evidence for the still controversial contribution of CELF1 to DM1, focusing on the muscle-specific overexpression of Bru-3, the *Drosophila* CELF1 family member, which contributes to pathogenic muscle defects similar to those seen in humans, including affected motility, fiber splitting, reduced myofiber length, and altered myoblast fusion [73]. Furthermore, the *Drosophila* model of DM1 that expresses 480 interrupted CTG repeats (i(CTG)480) was used to validate the appearance of an accelerated aging process exhibited in patients with DM1 [74]. Impaired locomotor activity and reduced lifespan compared with control flies lacking CTG repeats could be indicative of this and is confirmed by the identification of altered expression for markers of senescence like Dacapo (homolog of human p21CIP/p27KIP) and Psc (homolog of human BMI1) [74].

For DM2, the previous *D. melanogaster* models contributed to the identification and validation of new pathogenic mechanisms and potential therapeutic strategies but also of new, recently generated ones (reviewed in [53,54,75]). Thus, CCTG pathogenic length repeats expressed in the fly heart mimicked patient-like phenotypes [76]. Still intriguing is that DM2 follows a more favorable clinical course than DM1, despite both having very similar starting molecular mechanisms and phenotype types. This suggests that specific modifiers may modulate DM2 severity. In this regard, flies were involved in the characterization and validation of rbFOX1 RNA binding protein, the first factor specifically binding to expanded CCUG RNA repeats but not to expanded CUG RNA repeats [77]. Crossing flies overexpressing rbROX1 with different CTG and CCTG repeat expansion fly models was key to suggesting that the rbFOX1 protein competition with MBNL1 for binding to CCUG expanded repeats, partially releasing the latter from the repeat sequestration, was the potential mechanism linked to less severe phenotypes in DM2 [77]. The affordable genetic manipulation of flies has also been used to try to answer questions beyond repeat expansion involvement; for example, whether the dystrophic phenotype is also linked to a CNBP decrease [78]. For this purpose, dCNBP was depleted in muscles displaying ageing-dependent locomotor defects in flies. A correlation was observed with impaired polyamine metabolism and regulated through *dOdc* mRNA binding. This feature is also observed in muscles from DM2 patients, suggesting that the dCNBP function is easily evolutionarily conserved in vertebrates, with relevant implications for DM2 conditions. Finally, to investigate the toxicity of CCUG expansion and develop a useful model for drug screening, a new DM2 model was generated in *Drosophila melanogaster* with variable-length (CCUG)n repeats (*n* = 16 and 106). Through this new model, it has been observed that flies with (CCUG)_106_ display significant alterations in the external morphology of the eye and the underlying retina. Additionally, the expression of (CCUG)_106_ in developing retinas triggers a robust apoptotic response. These findings open up new avenues for studying the disease in *Drosophila* [54] (Table 1).

A similar strategy was followed for HDL2, caused by a CTG expansion in the *JPH3* gene, where the up- and downregulation of the single junctophilin (*jp*) gene was recently explored in a tissue-specific manner in Drosophila [32] (Table 1). The alteration of its expression levels, also previously connected to HD disease [32], produces a phenotypic spectrum characterized by muscular deficits, dilated cardiomyopathy, and neuronal alterations, establishing it as a good platform for evaluating the roles of the gene in HDL2 pathology but also for other vertebrate gene homologs, since further additional roles were uncovered.

For SCA31, the first *Drosophila* model was previously generated using expanded TGGAA repeats to explore toxic gain-of-function mechanisms [79]. These flies displayed significant RNA foci accumulation and interactions with RNA-binding proteins like TDP43, impacting both foci formation and observed degeneration features. Translation of the UGGAA repeat in all frames was detected as a poly-WNGME toxic protein, correlated with eye degeneration severity, suggesting contributions to neurodegeneration.

The use of FXTAS, a previous fly model based on the overexpression of 90 CGG repeats, established into the Fragile X (FXS) premutation range (55-200) [80], resulted in neuron-specific degeneration (reviewed in [81]). A few new studies using this model in recent years have highlighted the identification of modulators of CGG RNA toxicity. A fly targeted genetic screen using siRNA lines to downregulate genes in an FXTAS environment identified eight genes showing significant enhanced neuronal toxicity associated with CGG repeats, such as *Schlank* (ceramide synthase), *Sk2* (sphingosine kinase), and *Ras* (IMP dehydrogenase). Combined with a metabolic profiling in FXTAS mice, this validated that sphingolipid and purine metabolism are significantly perturbed in FXTAS pathogenesis [82]. A similar approach looking for genetic modifiers using a *Drosophila* model of FXTAS found 18 genes that genetically modulate CGG-associated neurotoxicity in flies, particularly *Prosbeta5* (*PSMB5*), since knockdown of *PSMB5* suppressed CGG-associated neurodegeneration in the fly but also in human cells [83]. Excitingly, a quantitative trait locus variant in *PSMB5*, *PSMB5rs11543947-A*, was found to be associated with decreased expression of PSMB5 and delayed onset of FXTAS in human FMR1 premutation carriers, thereby postulating this gene as important for the exploration of therapeutic strategies for FXTAS. An additional recent identification of players in FXTAS toxicity included DAP5 and SIMA factors [84,85]. Furthermore, flies carrying premutational CGG repeat length with the presence of neurodegeneration display restricted synaptic growth and a reduced number of synaptic boutons. This ultimately impairs synaptic transmission at the larval neuromuscular junctions, with a high percentage of boutons showing a reduced density of Bruchpilot protein, a key component of presynaptic active zones required for vesicle release [86]. FXTAS is one of the NREDs where RAN translation has been detected and is suggested to have a role as one of the toxicity processes involved in this pathology. Taking advantage of the fly models with an expanded CGG repeat in the context of the FMR1 50UTR, a candidate-based screen of eukaryotic initiation factors and RNA helicases was performed. This approach identified several modulators of RAN translation, including the DEAD box RNA helicase belle/DDX3X, the helicase accessory factors EIF4B/4H, and the start codon selectivity factors EIF1 and EIF5. Disruption of some of these factors inhibited FMR1 RAN translation in *Drosophila* in vivo and mitigated repeat-induced toxicity in *Drosophila* [87]. Finally, *Drosophila* helped to reposition in FXTAS the eIF5-mimic protein (5MP), a translational regulatory protein, proposed to reprogram non-AUG translation rates for oncogenes in cancer and with a suggested role in suppressing repeat-associated non-AUG (RAN) translation by a common mechanism dependent on its interaction with eIF3 factor. 5MP/Kra represses neuronal toxicity and enhances the lifespan in an FXTAS fly disease model [88].

#### 2.1.3. Use of Fly-Based Models in NREDs with Loss-of-Function Protein Mechanism

The expansion of a repeat tract can also trigger mechanisms leading to a partial or complete loss of function in specific proteins. Well-known examples are Fragile X syndrome (FXS) and Friedreich ataxia (FRDA), both pathologies previously modeled in flies [15,16] (Figure 1).

FXS, which can cause intellectual disability and has a suggested linked to autism spectrum disorders, is triggered by the transcriptional repression of *fragile X mental retardation gene 1* (*FMR1*) due to the hypermethylation of the CGG repeat motif when pathologically expanded (> 200 repeats) in the 5′-UTR of the gene. This mechanism causes the reduction or absence of encoding the fragile X mental retardation protein (FMRP) [89]. Several deletions and point mutations leading to the production of non-functional proteins have also been described [90]. The fly homolog *dfmr1* gene exhibits high sequence homology with human *FMR1* and its paralogs, *FXR1* and *FXR*2, but is most functionally related to FMR1 [91]. FXS fly models do not mimic the loss of the *dfmr1* gene by the hypermethylation mechanism but involve the generation of *dfmr1* loss-of-function stocks by the excision of transposable elements, EMS mutagenesis, or RNAi approaches (reviewed in [92]). These dFMR1 mutants recall several pathological symptoms of FXS patients like sleep problems, as well as deficits in memory, social interaction, and neuronal development [81].

It is accepted that the FMR1 gene has an important role influencing translation, but the associated mechanisms and the functional targets involved are still poorly understood. A recent study analyzed translation behavior in flies taking advantage of quiescent *Drosophila* oocytes, cells that, like neural synapses, depend heavily on translating stored mRNA. Wild type and FXS flies were compared after ribosome profiling. The results revealed that *dfmr1*’s role is to enhance, rather than repress, mRNA translation [93]. This increased activity preferentially targets large proteins. Close to fifty fly protein targets have been identified that are associated with human homolog genes already related with dominant intellectual disability and recessive neurodevelopmental dysfunction. Another interesting finding of this study was that stored oocytes lacking *dfmr1* usually generated embryos with severe neural defects, unlike stored wild type oocytes. These results suggest the importance of FMR1 in the appearance of FXS and potential connection with other autism spectrum disorders from a new perspective [93]. Additional studies have focused on identifying FMR1 targets. In this regard, the collapsing response mediator protein (CRMP) mRNA seems important in the abnormal circadian rhythm observed in FXS patients. In *dfmr1* mutant flies, the knockdown expression of *CRMP* fly homolog ameliorated the circadian defects and abnormal axonal structures of clock neurons observed in this model. A molecular analysis revealed the physical binding of FMR1 to CRMP, and altogether, these establish an essential role for CRMP in the circadian output in FXS [94]. A different target, also positively regulated by the *FMR1* gene, was discovered using FXS flies: the large A-kinase anchor protein (AKAP) Rugose, homolog of ASD-associated human neurobeachin (NBEA) [95]. This factor plays an important role in the central brain mushroom body (MB) circuit of the flies, where protein kinase A (PKA) signaling is necessary for learning/memory, and with *dfmr1* loss significantly reducing AKAP levels. A third important component in this pathway is F-actin, a well-established PKA-regulated protein important for cytoskeleton dynamics. In the FXS disease model, fly F-actin is aberrantly accumulated in the MB circuit, whose dysregulation seems important in FXS development [96]. In a related event also observed in FXS flies, cyclic adenosine monophosphate (cAMP) signaling via PKA is activated after exposure to *Drosophila* stress odorant (dSO). The dSO is a signal emitted when flies are subjected to electrical or mechanical stressors, which elicits an innate and robust avoidance behavioral response in wild-type *Drosophila* [97]. Additional FXS modulators recently identified from fly-based studies include the observation of higher *mad* transcript levels and pMad signaling after downregulation of *Drosophila* dFmr1 protein. Consistently, neuronal *dfmr1* and *mad* RNAi both decrease Prtp levels, which is related to the control of glial phagocytosis for correct circuit remodeling. These findings reveal a FMR1-dependent control pathway for neuron-to-glia communication in neurons [98]., A study in flies has shown that as a consequence of the *FMR1* gene role in translation modulation, dFmr1 loss also modulates the global metabolome, displaying reduced carbohydrate and lipid stores, with FXS flies displaying hypersensibility to starvation stress. *Dfmr1* seems to be a regulator of mitochondrial function, since electron micrographs of indirect flight muscles reveal striking morphological changes in their mitochondria. Together, these results illustrate the importance of *dfmr1* for the proper maintenance of nutrient homeostasis and mitochondrial function [99]. N6-methyladenosine (m6A) is the most prevalent modification found in mRNAs and regulates a variety of physiological processes through the modulation of RNA metabolism. This modification is particularly enriched in the nervous system of several species, and its dysregulation in *Drosophila* alters fly behavior and has been associated with neurodevelopmental defects and neural dysfunctions. An unbiased approach employed to identify m6A readers in the *Drosophila* nervous system led to the identification of the *Ythdf* gene. In flies, *Ythdf* directly interacts with *dfmr1* to inhibit the translation of key transcripts involved in axonal growth regulation [100]. Taking advantage of the rapid development of *Drosophila melanogaster* embryos, the regulation of mRNA localization to centrosomes was investigated, required for error-free mitosis and embryonic development. Interestingly from the perspective of FXS, a novel role for the dFmr1 protein was observed in the posttranscriptional regulation of a model centrosomal mRNA named centrocortin (*cen*). The study found that mistargeting of *cen* mRNA is sufficient to alter protein localization to centrosomes and impair spindle morphogenesis and genome stability [101]. *Drosophila* is also helping to shed light on how the *FMR1* gene is regulated at different RNA levels. The transcription product of *dfmr1* is a direct target of miR-315, a microRNA mainly expressed in the nervous system of flies. Flies overexpressing miR-315 showed pupation defects and reduced hatching rates, and the opposite situation (knockout) is embryonic lethal in flies. These observations indicate that miR-315 is a key regulator of the *Drosophila* nervous system and can be a factor to consider in FXS [102]. Additional miRNAs have also been predicted to regulate the *dfmr1* transcript, with specific miR-219 overexpression triggering a decrease in *dfmr1* expression in fly neurons, and *Drosophila* larvae showing morphological abnormalities at the neuromuscular junction (increased synaptic boutons and synaptic branches). This finding is consistent with some phenotypes observed in *dfmr1* mutants and suggests that miR-219 could be involved in FXS syndrome pathogenesis [103]. The use of novel open-source video annotation technology has allowed us to characterize new spontaneous-motor-behavior phenotypes of *Drosophila dfmr1* mutants. Researchers recorded individual 1-day-old *dfmr1* mutants, prior to the onset of aging, in small arenas. These flies displayed excessive grooming time, with increased bout number and duration (repetitive behavior). These phenotypes were rescued by inclusion of transgenic wild-type *dfmr1.* An accurate reading of courtship and circadian rhythm defects, already reported for *dfmr1* mutants, was established and provided a consistent readout to be used for drug discovery to identify compounds that reduce dysfunctional repetitive behaviors [104]. In a curious recent use, FXS flies were employed (along with wild-type *w^1111^* as controls) to investigate the neurodevelopmental impacts of bisphenol (BPA). BPA induced several aberrant phenotypes, such as hyperactivity in larvae, increased repetitive grooming behavior, and reduction of courtship behavior in adults, also impairing axon guidance in the mushroom body and disrupting neural stem cell development, all these in the wild-type genetic strain. Unexpectedly, BPA had null or insignificant impact in the above behaviors in FXS flies but did alter their behavioral and neuronal phenotypes, suggesting a gene–environment interaction between BPA and the *dfmr1* that deserves further research [105,106].

Friedreich ataxia (FRDA), the most common inherited ataxia, is characterized by progressive neurodegeneration and cardiomyopathy, the latter being the most common cause of death in patients. FRDA is caused by the transcriptional silencing of the FXN gene, which codes for frataxin, a mitochondrial protein involved in iron–sulfur cluster biosynthesis. The expansion of the GAA tract in intron 1 elicits transcriptional silencing by the formation of non-B DNA structures (triplexes or sticky DNA), a persistent DNA x RNA hybrid, or heterochromatin. Despite being a mitochondrial protein, little is known about the influence of frataxin depletion on homeostasis of the cellular mitochondrial network. Using a RNAi model to deplete endogenous *fh* gene in flies, a genetic screen was performed to analyze genetic interactions between genes controlling mitochondrial homeostasis and *Drosophila* frataxin. The most significant finding was the silencing of *Drosophila* mitofusin (*Marf*) gene as a suppressor of FRDA phenotypes in glia. This result suggests that the protection mediated by *Marf* knockdown in FRDA glia is mainly linked to its role in mitochondrial–ER tethering and not to mitochondrial dynamics or mitochondrial degradation [107]. It is important to mention that fly models for FRDA initially only involved depletion of the *fh* homolog fly gene. Recently, a more accurate fly model was developed, named fh-200GAA, and generated using CRISPR/Cas9 technology to better mimic the activity of the human gene. These flies contain the insertion of a GAA expansion of approximately 200 units in the homolog frataxin (*fh*) gene of the fruit fly in a similar position as in the human gene, in the first intron of the *fh* gene, and flanked by sequences from the human intron of FXN (Table 1). Characterization of the line displayed developmental delay and lethality associated with decreased frataxin expression. Using genetic tools, preadult lethality was bypassed, limiting *fh* loss to after the developmental period. Adult frataxin-deficient flies are short-lived and present strong locomotor defects. Further genomic studies on this line by RNA-seq identified deregulation of several genes involved in amino acid metabolism and transcriptomic signatures of oxidative stress. One interesting observation was the progressive increase in *Tspo* expression, fully rescued by adult frataxin expression. Thus, *Tspo* expression constitutes a molecular marker of disease progression in this new fly model and might be of interest in other animal models or in patients. Overall, this FRDA *Drosophila* model provides a promising experimental platform for investigating the underlying mechanisms of Friedreich ataxia [55]. Linked to the previous results, it has been reported that frataxin serves as an allosteric regulator for cysteine desulfurase, the enzyme that provides sulfur for [2Fe–2S] cluster assembly. An X-ray crystallography and nuclear magnetic resonance (NMR) spectroscopy study indicated that bacterial, *Drosophila melanogaster*, and human frataxin proteins are structurally similar, confirming the high conservation at the functional level and utility of FRDA fly models [108]. The compromised function of the reticulum–mitochondria associated membranes (MAMs) observed in a FRDA cellular model (inter-organelle structures involved in the regulation of essential cellular processes, including lipid metabolism and calcium signaling) was improved by promoting mitochondrial calcium uptake in flies via genetic overexpression of the mitochondrial Ca 2+ uniporter (*MCU*) gene, previously proven to alter mitochondrial calcium content in flies and sufficient to restore several defects present in the FRDA mutant flies. Remarkably, the results of this study represent the first time frataxin has been reported as a member of the protein network of MAMs, implicated in endoplasmic reticulum–mitochondria communication, and suggest a new role beyond mitochondrial defects for this protein, pointing to MAMs as novel therapeutic candidates to improve patient condition [109].

#### 2.1.4. Use of Fly-Based *c9orf72* ALS/FTD Models

The G4C2 hexanucleotide repeat expansion in intron 1 of the chromosome 9 open reading frame 72 (*c9orf72*) gene is the most prevalent factor linked to amyotrophic lateral sclerosis (ALS) and frontotemporal dementia (FTD) neurological disorders [110], but doubts still remain as to the importance of the different disease-based molecular mechanisms suggested (Figure 1). These include: (i) a repeat RNA gain-of-function, like in DM1 or FXTAS, also accompanied by the formation of intranuclear RNA foci and the sequestering of key RNA-binding proteins (RBPs); but also, (ii) the unconventional, repeat-associated non-ATG (RAN) translation, leading to the production of toxic poly-dipeptides, collectively termed dipeptide repeat proteins (DPRs), with up to five distinct molecules being generated (poly-PA, poly-PR, poly-GA, poly-GP, and poly-GR) [111], and finally, the less explored: (iii) loss of gene function [112]. Despite the absence of *c9orf72* homolog in flies, this pathology has been by far the most intensively studied NRED using *Drosophila melanogaster* during recent years. This provides the rationale behind dedicating a separate section specifically to fly-based studies in ALS/FTD, and achieving valid models is a key requirement for widening our understanding in this area. 

The complexity of the repeat motif has limited the generation of in vivo models expressing long *c9orf72* related DPRs, and the repeat length is usually only a few hundred repeats. Regarding the abovementioned G4C2 repeat expansion, new fly-based models expressing DPRs over 1000 repeat units in length have recently been generated. Their characterization demonstrated that each model expressing a specific DPR exhibited a unique, age-dependent, phenotypic and pathological profile. Furthermore, for the first time, studies focused on more realistic situations in which the co-expression of specific DPR combinations is conducted in humans. In flies, simultaneous DPR expression led to age-dependent phenotypes, some not observed through the expression of single DPRs [56] (Table 1). The generation of specific DPRs is gaining importance as a source of ALS/FTD-based toxicity. Different studies have shown that arginine-rich (poly-GR and poly-PR) DPRs are the most toxic. However, polyGlycine-Alanine (poly-GA) is the most abundant DPR in patient brains. As previously mentioned, several models have been generated expressing DPRs of physiologically and pathologically relevant lengths for the disease. Whether repeat length affects the toxicity of DPRs has not been systematically assessed. Fly stocks expressing, for example, different poly-GA lengths in adult neurons showed that expression of poly-GA100 and poly-GA200 caused only mild toxicity, in contrast with neuronal expression of poly-GA400. This last event drastically reduced climbing ability and survival of flies, indicating that long poly-GA DPRs can be highly toxic in vivo [113]. A proteomics analysis of fly brains also showed a repeat-length-dependent modulation of the brain proteome, with poly-GA400 causing earlier and stronger changes than shorter GA proteins [113]. However, the mechanisms underlying their expression are not fully understood. A *Drosophila* genetic screen identified ribosome-associated quality control (RQC) factors as potent modifiers of poly(GR)-induced neurodegeneration. This result is connected to the suggested mechanism of poly(GR)-encoding sequences causing translational stalling, and thus generating ribosome-associated translation products, sharing molecular signatures with canonical RQC substrates. Zfp106, an RNA-binding protein essential for motor neuron survival, suppressed neurotoxicity in a *Drosophila* model of *c9orf72* ALS by binding and causing a conformational change in the RNA G-quadruplexes formed by GGGGCC repeats, thus inhibiting RNA foci formation and reducing DPR levels. However, what is happening at the cellular level? Different studies in flies are pointing to the disruption of nucleocytoplasmic transport (NCT) as one of the most relevant issues. It has been characterized that the expression of G4C2 repeats in *Drosophila* neurons triggers the translocation of nucleoporins into the cytoplasm and further proteasome-mediated degradation. Specifically, the upregulation of nuclear *ESCRT-III/Vps4* gene expression seems to be involved in this mechanism since its knockdown normalizes the transport process and suppresses GGGGCC-mediated neurodegeneration [114]. Previously, in a subgroup of patients with ALS/FTD, a transport deficit was suggested due to binding of the G4C2 repeat expansion to Ran-activating protein (RanGAP) at the nuclear pore, which accumulates in the cytosol. Using *c9orf72* flies, the pathological effects defined by the co-localization of RanGAP and nuclear pore proteins (Nups) was reversed by overexpression of the molecular chaperone sigma-1 receptor (Sig-1R) but not the Sig-1R-E102Q mutant, probably by stabilizing the whole structure. Interestingly, Sig-1R directly binds (G4C2)-RNA repeats [115]. In addition, the partial depletion of SRSF1, directly involved in nuclear export processes, as well as of its interaction with related factors, such as the human KCNN1–3 (*Drosophila* SK) voltage-gated potassium channel orthologs, were recently identified playing neuroprotective roles [116,117]. Furthermore, the upregulation of proteins involved in the endoplasmic reticulum (ER) to Golgi trafficking and the downregulation of proteins involved in insulin signaling were detected. Thus, the fly-based experimental downregulation of Tango1, increasing insulin signaling, partially rescued GA400 toxicity [118]. The marked accumulation of poly(A) mRNAs in the cell nuclei is providing additional clues about the presence of a major dysregulation of the nucleocytoplasmic transport, contributing to the pathogenesis of ALS. Functional studies in *Drosophila* indicated that *c9orf72* toxic species affect the membrane trafficking route by altering the function of the ADP-ribosylation factor 1 GTPase activating protein (ArfGAP-1), finally dissociating coat proteins from Golgi-derived vesicles [119]. Importantly, poly-GR expression in various neuronal populations in *Drosophila* induces a variety of pathological phenotypes like axonal degeneration and synaptic dysfunction, mainly in glutamatergic neurons, including motor neurons, triggering high levels of extracellular glutamate and intracellular calcium, motor deficits, and shortened life span, thus indicating a cell-autonomous excitotoxicity mechanism [120]. Recently, DPRs have also been found in the sleep-related neurons of patients, indicating a role in ALS-associated sleep disruptions. Poly-GA or poly-PR DPRs were expressed in male *Drosophila melanogaster,* with only the latter ones causing sleep disruptions. This study validates the use of flies as an in vivo model system for exploring the roles of DPRs in perturbing the underlying molecular mechanisms in sleep regulation [121].

In terms of RNA toxicity, the effects of expanded G4C2 repeat RNA levels and how to reduce them are also under investigation in *Drosophila*, with new neurodegeneration-related pathways already identified. In this regard, RBPs are suggested to play an important role, as confirmed by several fly-based studies. The RBP matrin-3 (MATR3) has been found colocalizing with G4C2 RNA foci in patient tissues, exhibiting perturbed subcellular distribution in neurons. Several studies, including those using fly *c9orf72* models, have suggested that MATR3 genetically modifies the neuropathology and pathobiology of *c9orf72* ALS through modulating the RNA foci and RAN translation [122]. Another suggested new pathological feature is the in vivo nuclear accumulation of a cytoplasmic RBP (Staufen) in neurons expressing poly-PR [123]. A different hypothesis suggests that the nonsense-mediated decay (NMD) pathway, involving RNA regulation processes, is linked to the G4C2 hexanucleotide repeat expansion. However, although a study characterized that overexpression of UPF1 (a protein involved in NMD) significantly reduced poly-GP abundance and thus the severity of known neurodegenerative phenotypes in a *c9orf72* fly model, the amount of repeat RNA was not altered, unravelling the protective role from the NMD pathway [124]. The elevated expression of a different subset of human RNA-binding proteins that bind to GGGGCC repeat RNA has been also characterized, including hnRNPA3, IGF2BP1, hnRNPA2B1, hnRNPR, and SF3B3, here able to reduce the level of the toxic RNA, resulting in the suppression of neurodegeneration. These results provide evidence for the therapeutic potential of the repeat RNA-lowering approach mediated by endogenous RNA-binding proteins for the treatment of c9-related ALS/FTD [125].

Higher than normal levels of the topoisomerase 2 (*Top2*) gene (linked to key DNA processing mechanisms) were detected in *Drosophila c9orf72* models, as well as in the brains of *Sod1G93A* model mice. This suggests that elevated levels of topoisomerases may be involved in a pathway common to the pathophysiology of distinct ALS forms [126]. In relation to the latter, an RNA-seq analysis of poly(GR) toxicity in *Drosophila* found several antimicrobial peptide genes, such as metchnikowin (*Mtk*) and heat shock protein (*Hsp*) genes, to be activated. Curiously, *Top2* regulates poly(GR)-induced upregulation of *Hsp90* and *Mtk* [127]. Together, these results suggest potential relevance for the repeat DNA structure and for identifying novel therapeutic targets for *c9orf72*-ALS/FTD. In this regard, the FUS ribonuclear binding protein suppresses RAN translation and neurodegeneration in an RNA-binding activity-dependent manner [128]. These results reveal a previously unrecognized regulatory mechanism of RAN translation by G-quadruplex-targeting RBPs, providing therapeutic insights for c9-ALS/FTD and other repeat expansion diseases [128]. Likewise, a CRISPR-Cas9 screen for modifiers of DPR protein production in human cells revealed that DDX3X, an RNA helicase, suppresses the repeat-associated non-AUG translation of GGGGCC repeats. DDX3X directly binds to (GGGGCC)n RNAs but not antisense (CCCCGG)n RNAs. Helicase activity is essential for translation repression, and in *Drosophila*, DDX3X reduction enhanced G4C2-mediated toxicity [129]. Similarly, senataxin (*Setx*) helicase has been validated as a modulator of DPR toxicity in flies. A generation of *Setx* fly lines were used to be crossed with flies expressing either (G4C2)58 repeats or glycine-arginine-DPRs. The dramatic suppression of disease phenotypes in (G4C2)58, GR(50), and GR(1000) *Drosophila* models uncovered a striking re-localization of GR(50) out of the nucleolus in flies co-expressing *Setx* [130]. Additional fly-based studies have also provided promising pathways and genetic disease modifiers, such as Lilliputian (*Lilli*, with four homologues in mammals), which suppresses poly(GR) toxicity by specifically downregulating the transcription of GC-rich sequences in *Drosophila*, also exhibiting alteration of TDP-43 protein localization, collectively sustaining additional mechanisms underlying ALS/FTD pathologies [131]. The association between apoptosis and autophagy in the context of disease related to *c9orf72* repeat expansion is a key area of research that has received significant attention in recent studies. Imaging experiments in flies revealed that expanded G4C2 repeat expression significantly affects autophagosome biogenesis, specifically at synaptic terminals. This suggests that G4C2 repeat expansion has a direct impact on autophagosome formation at critical sites for neuronal function, which could contribute to the observed autophagy dysfunction in *c9orf72*-related ALS/FTD [118]. A screening approach in *Drosophila* identified *p62*, a key regulator of autophagy, as a potent suppressor of neurodegeneration caused by G4C2 expansion. p62 is increased and forms ubiquitinated aggregates due to decreased autophagic cargo degradation, partially caused by cytoplasmic mislocalization of Mitf/TFEB, a key transcriptional regulator of autophagolysosomal function. These data suggest that the G4C2 repeat expansion impairs Mitf/TFEB nuclear import [132]. Additionally, the observation of aberrantly accumulated Tau protein together with reduced autophagy levels seems to be linked to neurodegeneration processes in ALS/FTD [133]. Additionally, in a different unbiased, large-scale screen in *Drosophila* flies expressing (G4C2)49, the CDC73/PAF1 complex (PAF1C), a transcriptional regulator of *RNAPII*, was identified as a suppressor of repeat-associated toxicity. Interestingly, dPAF1C components dPaf1 and dLeo1 appear selective for transcription of long, toxic repeat expansions but not shorter, non-toxic expansions, and PAF1C is upregulated upon expression of (G4C2)30+ in flies and mice [134]. The activation of a specific transcriptional program, mediated by the p53 transcription factor, is also associated with the neuronal degeneration in *c9orf72* mouse and fly models. Ablation of *p53* completely rescues neurons from degeneration and increases survival in these models, suggesting a crucial role for *p53* in disease pathogenesis. This evidence supports the idea that apoptosis, mediated by *p53*, also significantly contributes to neurodegeneration in *c9orf72*-related ALS/FTD [135].

It has been observed that *c9orf72* DPRs also disrupt the proteasome and perturb proteolytic activities in human and fly brain tissues. Co-immunoprecipitation in *D. melanogaster* demonstrated that poly-PRs bind strongly to the proteasome, and the impairment is further shown by the accumulation of ubiquitinated proteins, along with lysosomal accumulation and hyper-acidification [136]. Lipid alterations in the brain are also well-documented in disease and aging, but our understanding of their pathogenic implications remains incomplete. The use of time-of-flight secondary ion mass spectrometry (ToF-SIMS) with G4C2 repeat expansion fly models distinguished that expressing them in the brain elevates the levels of fatty acids, diacylglycerols, and ceramides during the early stages of disease progression, preceding motor dysfunction. An RNAi-based genetic screening targeting lipid regulators identified that reducing fatty acid transport protein 1 (FATP1) and Acyl-CoA-binding protein (ACBP) alleviates the retinal degeneration of flies caused by G4C2 repeat expression and also substantially restores G4C2-dependent alterations in lipid profiles. Significantly, *fatp1* and *acbp* expression are upregulated in G4C2-expressing flies, implying their contribution to lipid dysregulation [137]. Finally, fly-based results are suggesting additional non-neuronal pathogenic features in ALS/FTD, observed when poly-GR enters mitochondria in muscle cells and impairs mitochondrial inner membrane structure, ion homeostasis, mitochondrial metabolism, and muscle integrity. Similar mitochondrial defects are observed in patient fibroblasts [138].

### 2.2. New NREDs Modeled in Drosophila Melanogaster 

#### 2.2.1. NOTCH

The recent research has unveiled new CGG repeat extensions at different loci contributing to a variety of neuromuscular and neurodegenerative disorders with similar symptoms and pathological features [139,140]. Among these is neuronal intranuclear inclusion disease (NIID), characterized by extensive intranuclear inclusions in the nervous system and various internal organs, originating from CGG repeat expansions in the 5’ untranslated region (5’UTR) of the human-specific *NOTCH2NLC* gene [141], involved in cortical neurogenesis in humans. Studies have revealed that expansions ranging from 60 to 500 are linked to NIID, which can manifest at different stages of life: infancy, adolescence, or adulthood [142]. Clinically, NIID presents with a variety of symptoms including dementia, muscle weakness, peripheral neuropathy, cerebellar ataxia, parkinsonism, seizures, and encephalitic episodes. The molecular mechanisms underlying NIID pathogenesis remain unclear due to the scarce number of genetic models. The current mouse models for NIID involve overexpression of *NOTCH2NLC* with artificial pathogenic CGG repeat expansions in the brains of adult mice through retro-orbital adeno-associated virus (AAV) injection or in utero electroporation in the mouse neocortex [143]. These models make it challenging to delineate the specific effects of pathogenic CGG repeat expansions at different ages and in different types of tissues. Recently, the first transgenic *Drosophila* model expressing uN2CpolyG in multiple systems was established. Progressive neuronal cell loss, locomotor deficiency, and shortened lifespan were characterized in the flies. The specific expression of uN2CpolyG in fly eyes led to progressive retinal degeneration. To investigate the pathogenic function of uN2CpolyG in the fly model, researchers used the UAS-Gal4 system to express GFP proteins in control (uN2C-GFP) and NIID (uN2CpolyG-GFP) flies. The expression of uN2CpolyG resulted in mild rhabdomere loss at day 5, and severe ommatidial degeneration at day 30. Additionally, intranuclear inclusions were observed in flies expressing uN2CpolyG but not in those expressing the control uN2C protein (Table 1). These findings underscore the value of the *Drosophila* new model in exploring the disease and the pathogenic role of the uN2CpolyG protein in NIID-linked neurodegeneration [57].

#### 2.2.2. FAME 

Familial adult myoclonic epilepsy (FAME) is a neurological disorder caused by the intronic expansion of TTTTA/TTTCA. The pathological mechanism remains unknown, but it is understood that the six genes harboring FAME-associated expansions encode proteins with very different subcellular localizations, suggesting that the pathological mechanisms may be independent of gene type. However, RNA toxicity associated with these repeats has been observed [144]. The first studies have been conducted with *Drosophila* to elucidate the role of genes harboring repeat expansions. One such study focuses on the *YEATS2* gene, whose expansion leads to FAME. This study employed pan-neuronal silencing of *dYEATS2* in *Drosophila*, evaluating the resulting molecular and behavioral effects. Reduction in *dYEATS2* expression resulted in decreased tolerance to acute stress, altered locomotion, abnormal social behavior, and reduced motivated activity. It also negatively affected the expression of the tyrosine hydroxylase (TH) gene, decreasing dopamine biosynthesis. The seizure-like behaviors induced were rescued with L-DOPA. This suggests that YEATS2 plays a role in regulating acute stress responses, locomotion, and complex behaviors and that its haploinsufficiency could be involved in FAME4 [58] (Table 1). The next step should involve generating models with the specific repeat expansion causing FAME.

## 3. NREDs and Fly-Based Therapeutic Programs

Fly-based models of human diseases provide several unique features for applied research, including a few that can be used to identify and/or validate drug candidates. These include the development of low- to high-throughput drug screens, as well as approaches for hits validation and target discovery using several different NRED fly models.

For toxic RNA models, the cardiac DM1 model expressing 480 CTGs in the heart was used to validate daunorubicin hipochloride, identified from an in vitro screening of ∼6500 compounds from the Strasbourg Academic and Prestwick libraries and using the ability to bind to the CUG repeats as a target readout. The validation processes in flies included daunorubicin evaluation for cardiac function assessment in 7-day-old flies, fed with the compound, dissected to expose the beating heart in artificial aerated hemolymph, with video record taken and analyzed by SOHA software (http://www.sohasoftware.com/). Flies fed with daunorubicin showed significant improvement in several cardiac disease parameters, including heart contractility, compared to DMSO-fed model flies. Furthermore, the survival phenotype was increased in DM1 flies fed with daunorubicin, increasing from under 30 to 40 days, close to the 47 days of median survival in control flies with no CTG repeats [145]. In a different DM1-based study, a virtual screening was carried out based on the ability of small molecules to bind to the CUG hairpin structure. Different hits including pyrido[2,3-d] pyrimidines and pentamidine-like compounds were identified, and some were validated upon assessing improvement of the impaired locomotion phenotype in the climbing test present in DM1 flies when CTGs are expressed in the muscle [146]. A different approach used novel spliceosensor DM1 flies, involving the transgenesis of human DM1-linked splicing events for a large-scale pharmacological screening of >20,000 small molecules to identify boldine, a natural alkaloid, which was validated after its ability to modify disease phenotypes was observed in several DM1 models, including the reduction of nuclear RNA foci in DM1 cell lines, and noteworthy anti-myotonic activity in the HSA^LR^ mouse model. These results position boldine as an attractive new candidate for therapy development in DM1 [147]. The treatment of 480 CTG flies with senolytic compounds reverses the accelerated aging processes observed and is linked to the identification of senescence markers in DM1, introducing a novel therapeutic opportunity for patients [74]. DM2 flies have also recently been used for a screening of 3,140 FDA-approved drugs, in this case, using lethality and locomotion phenotypes after expression of 720 CCTG repeats in the muscle. Interestingly, four identified hits shared predicted targets in the TGF-β pathways, and validation steps included the use of specific inhibitors through this pathway that ameliorated climbing defects, crushed thoraxes, structure, and organization of muscle fibers in DM2 flies [148]. Turning to FXTAS, a well-established *Drosophila* model was recently used for a high-throughput chemical screen of 3,200 small molecules. The results showed that compound NSC363998 was able to significantly suppress the neurodegeneration caused by CGG repeats. The predicted targets of a NSC363998 derivative are isopeptidases. These targets connect the neddylation pathway, an analogous process to ubiquitination, although relying on its own enzymes to modulate the neurotoxicity caused by CGG repeats, as a potential therapeutic target for FXTAS [90]. Finally, also in FXTAS, the pharmacologic inhibition of serine/arginine protein kinases (SRPKs) in flies has been able to inhibit RAN translation of CGGs but also of GGGGCC repeats (associated with *c9orf72*), suppressing toxicity in both disease models and also opening a novel therapeutic target in RNA toxic diseases [85].

Regarding the development of recent therapeutic-like studies using flies for NREDs with poly-protein toxicity, a previous screening identified AF2 modulators for their ability to rescue toxicity in a *Drosophila* model of SBMA. Two of them, tolfenamic acid (TA) and 1-[2-(4-methylphenoxy)ethyl]-2-[(2-phenoxyethyl)sulfanyl]-1H-benzimidazole (MEPB), rescued lethality, locomotor function, and neuromuscular junction defects in SBMA flies and were further validated in a novel mouse model of SBMA [149]. Aside from this, reducing proteasome activity using the inhibitor MG132 was able to improve muscle function in the OPMD *Drosophila* model and provides proof-of-concept data indicating the modulation of proteasome activity as an attractive pharmacological approach for OPMD patients [52]. The oral treatment of OPMD flies with icerguastat decreased muscle degeneration and PABPN1 aggregation, also revealing the major contribution of ER stress in OPMD pathogenesis [150]. A similar result was observed from the treatment with certain anti-prion molecules such as flunarizine, azelastine, duloxetine, ebastine, loperamide, and metixene, alleviating phenotypes due to PABPN1 nuclear aggregation and emphasizing the therapeutic potential of anti-PFAR drugs, which are able to target protein folding, for neurodegenerative and neuromuscular proteinopathies [151]. The recent studies have explored various therapeutic strategies also in HD using fly models. In this way, supplementation with the natural compound *Rhodiola rosea* in adult flies expressing 93Qs resulted in significant improvements in longevity, locomotion, and neurodegeneration, despite the observed toxic effects in larval stages, through its capacity to inhibit the mTOR pathway and induce autophagy. These findings suggest therapeutic potential for this plant in disease treatment [152]. In other research, the discovery of increased copper and iron concentrations in the striata of post-mortem human HD brains, and that accumulation of mutant HTT protein can interact with copper, raises a question regarding the underlying HD progressive phenotypes due to copper overload. Using a *Drosophila* model of HD, researchers showed that copper induces dose-dependent aggregational toxicity and enhancement of Htt-induced aggregation and neurodegeneration and consequently alters autophagy in the brain. Treatment with the copper chelator D-penicillamine (DPA) through feeding significantly decreases β-amyloid aggregates in the HD pathological model [153]. Additional anti-HD small molecules recently found or validated using fly models include curcumin [154,155], ginseng extract [156], fullerenols [157], 6-azauridine (6-AZA) [158], teglicar [159], gossypol [160], AQAMAN [161], and meldonium [162], but also other type of molecules like MS3 and shortened MS3–33 and MS3–17 aptamers (oligonucleotides) able to selectively bind and modulate the activity of mutant huntingtin (mHTT) by a distinctive propensity to form complex G-quadruplex structures [163,164], peptides contained in nanoparticles [165,166], and natural extracts like green tea infusion [167].

This review has previously highlighted the high level of complexity of the *c9orf72* ALS/FTD disease, with distinct mechanisms of action present. A recent study has shown the importance of (fly) model selection when several are available, as is the case for the evaluation of drug candidates [168]. The authors investigated the potential for specific therapeutic candidates, with different mechanisms of action, in mitigating functional features in several G4C2-expressing fly lines. These were TMPyP4, destabilizing G-quadruplexes, potentially affecting the stability of G4C2 repeats and reducing both RNA and protein toxicity; PJ34, inhibitor of poly (adenosine diphosphate-ribose) polymerase and considered for attenuating protein toxicity; and KPT-276, inhibiting the nuclear export of proteins and RNA and thus potentially affecting G4C2-related pathologies through modulating nucleocytoplasmic trafficking. The results published demonstrated dissimilarities in response among the fly models used with the same compound. Therefore, the accurate characterization of models for the different mechanisms is crucial for interpreting the therapeutic potential proposed for a specific compound [168]. Nonetheless, promising results have been achieved in the field of drug development in recent studies with G4C2 fly models. One interesting feature evaluated at the therapeutic level concerns the way to remove the structures formed by DPRs. It is known that Poly-GR/PR, the most toxic one, binds to nucleotides and interferes with transcription. A *Drosophila* model helped to validate a sulfated disaccharide found to bind to poly-GR/PR, which was able to rescue its shortened life span and defective locomotion [169]. Similarly, some natural marine products (chrexanthomycin, A, B and C) selectively bind to the G-quadruplex structures formed by DNA/RNA G4C2 motifs, and in (G4C2)29 expressing *Drosophila*, two of them significantly rescue eye degeneration and improve locomotor deficits [170]. Recently, crystal structures of DNA and RNA–DNA hybrid duplexes with the GGGCCG region as a G4C2 repeat motif were solved, displaying the formation of unusual bending structures, used to better define the binding of designed molecules. Chro-metal complexes therefore inhibit neuronal toxicity and suppress locomotor deficits in a *Drosophila c9orf72* model by binding to these structures, and the approach represents a new targeted direction for drug discovery against ALS and FTD diseases by targeting G4C2 repeat motif DNA [171]. *Drosophila* models of poly-GR overexpression have been used for the first time to assess the therapeutic ability of oligonucleotide-based and cell-penetrating peptide molecules, achieving a moderate but significant reduction in severity of DPR toxicity observed as dependent degeneration, the appearance of necrotic patches, and locomotion deficits [172,173]. Similarly, from a cell-based high-throughput drug screen, some compounds known to boost protein kinase A (PKA) activity increased DPR levels. The strategy was to evaluate PKA inhibitors, with some rescuing pathological phenotypes in a *Drosophila* model of c9ALS/FTD [174]. These results suggest new druggable pathways modulating DPR levels. Further evidence showed that chemical inhibition of the *Top2* gene to downregulate its high expression in fly models alleviated repeat expansion-mediated neurotoxicity [126]. Similarly, proteasome perturbation by the presence of DPRs triggering accumulation of ubiquitinated proteins can be rescued by a small-molecule proteasomal enhancer [137]. Regarding the phenomenon of lipid dysregulation observed in G4C2 flies, nonanoic acid (NA) and 4-methyloctanoic acid (4-MOA) have been shown to ameliorate impaired motor function in *c9orf72* larvae and improve NMJ degeneration, although their mechanisms of action are not identical [137,175]. NA modified postsynaptic glutamate receptor density, whereas 4-MOA restored defects in presynaptic vesicular release. Finally, manipulation of several mitochondrial components altered in muscle cells, together with pharmacological restoration of ion homeostasis with nigericin, effectively rescues the novel mitochondrial pathology and disease phenotypes observed in flies expressing G4C2 expansions [138]. Taken together, these findings underscore the utility of *Drosophila* models in unraveling the complex molecular mechanisms involved in neurodegenerative diseases and identifying potential therapeutic targets that could alleviate pathology in patients with *c9orf72*-related ALS and FTD. Finally, the pharmacological inhibition of serine/arginine protein kinases (SRPKs) directly inhibits RAN translation of CGG and G4C2 repeats associated with C9orf72 in flies [117].

Fly models for NREDs with loss-of-function protein have been used for exploring therapeutic molecules. Acamprosate, approved in 2004 to treat relapse from alcohol withdrawal, used in off-label studies in FXS, was evaluated in *Drosophila dfmr1* null mutants and wild-type larvae. The treatment with the repurposed molecule partially or completely rescued all the FXS phenotypes analyzed, except crawling behavior, at the high dose. Low doses of acamprosate, however, did not affect synapse number at the NMJ but could rescue NMJ overgrowth, locomotor defects, and cbp53E mRNA expression. This dual nature of acamprosate suggests multiple molecular mechanisms may be involved in acamprosate function depending on the dosage used. Nonetheless, improved understanding is needed of the molecular mechanisms involved with different doses of this drug [176]. As previously mentioned, *dfmr1* mutant flies display a significant defect in responding to dSO, but several drugs regulating both cAMP and cyclic guanosine monophosphate (cGMP) levels significantly improved it [97]. A similar approach was used to identify the mitogen-activated protein kinase (MAP3K) Wallenda/dual leucine zipper kinase (DLK) as a critical target of FMR1. dFmr1 binds Wallenda mRNA and is required to limit its protein levels. The pharmacological inhibition of Wallenda in larvae suppresses dFmr1 neurodevelopmental phenotypes, while adult administration prevents dFmr1 behavioral defects, now suggested as a candidate therapeutic intervention for the treatment of FXS [177]. Regarding FRDA, ER stress chemical modulation, as observed in OPMD flies with icerguastat, was reduced after chemical treatment with mitofusin and was able to ameliorate the effects of frataxin deficiency in three different fly FRDA models. This work could define a new pathological mechanism in FRDA, suggesting ER stress as a therapeutic target in FRDA [107]. Specifically, when evaluating frataxin inactivation in the heart of flies, which triggers heart dilatation and impaired systolic function, methylene blue (MB) and paclitaxel were highly efficient to prevent these cardiac dysfunctions [178]. These two molecules were identified from the fly FRDA model after screening the Prestwick Chemical Library, comprising >1,200 compounds. This study is the first drug screening of this scope performed in vivo on a *Drosophila* model of cardiac disease. Thus, it also brings the proof of concept that cardiac functional imaging in adult *Drosophila* flies is usable for medium-scale in vivo pharmacological screening, with potent identification of cardioprotective drugs in various contexts of cardiac diseases [178]. Additionally, treatment with antioxidants, such as N-acetyl cysteine (NAC), was able to recover survival and locomotor phenotypes linked to elevated oxidative stress in FRDA flies [55,109].

## 4. Future Challenges

The outstanding response and large number of advances provided by the various models generated for NREDs in *Drosophila melanogaster* organism advocates, as a primary future challenge, for the development of new transgenic fly-based models that will contain still never modeled disease-related repeat tracts. Examples would include repeat expansions such as AAGGG for CANVAS, GGCGCGAGC for HMN, or CCCTCT for XPD, among others, all rare diseases of recent discovery [5]. Thus, the first in vivo tools would be available to decipher the way to target them efficiently.

As commented throughout the previous sections, distinct strategies could be utilized to generate these models, from removal or overexpression of the affected gene or simple genomic insertion of the repeat tract, to insertion of the expanded repeat tract in the same (*Drosophila* homolog) gene or insertion of the repeat in the same human genomic background by adding human flanking sequences. During recent years, diseases such as HD and SCAs have benefited from well-established fly models, yet many other NREDs (such as *c9orf72* or DMs) still lack more robust fly models that can better mimic the human situation. Gaps include the evaluation of repeat interruptions, repeat methylation status, and the identification of roles for repairing or replicating genes, which has still not been approached in flies. By creating fly models for such underexplored conditions, researchers can gain a better understanding of the pathological mechanisms and dynamics of repeat instability within a simpler and highly manipulable system.

Altogether, the existing fly models are crucial for understanding key disease mechanisms, defining new disease players, and searching for valid therapeutic options, but among the data still not acquired, and also not systematically approached, is the reproduction of the key high levels of repeat instability (somatic, intergenerational) observed in patients and only reproduced in some mouse NRED models [179]. In flies, only one published study shows intergenerational repeat instability in CAG repeat models [180]. It would be encouraging to use novel long-read sequencing techniques (PacBio, Oxford Nanopore) to study repeat instability in already existing fly models to elucidate whether repeat expansions or deletions have been observed in the models currently being used. Likewise, the design of crossing protocols to include the evaluation of repeat instability levels in fly models is necessary.

In conclusion, our review highlights several new players involved in NRED toxicity being discovered through fly-based models, with the identification of novel therapeutic targets and drug candidates becoming a central focus. Taking advantage of this review, we want to emphasize the high, still not fully realized potential of flies in the field of drug development in human disease. We found examples of solid innovation utilizing flies via easy to use high-throughput in vivo genetic or chemical screenings [137,147] but also testing novel type of compounds, such as antisense oligonucleotides, that can target repeat expansions very specifically [163,172]. Finally, methodical studies are warranted to determine the ingestion processes of this type of molecule in flies.

## Figures and Tables

**Figure 1 ijms-25-11794-f001:**
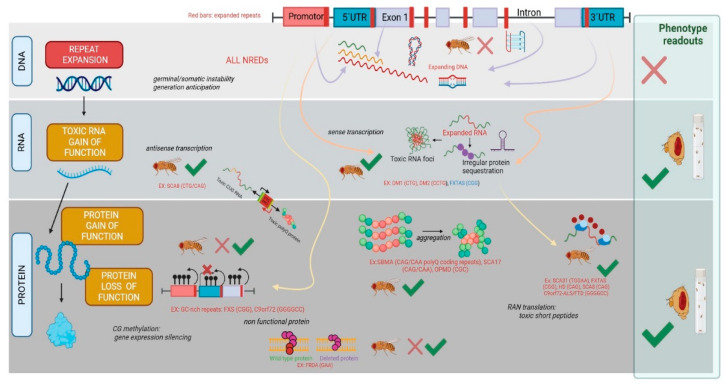
**Nucleotide repeat expansion disease pathways and mechanisms and fly modeling.** In NRED studies, several disease-linked pathways and mechanisms have been identified, all starting with (i) uncontrolled DNA repeat expansion in a specific gene, spotting distinct possibilities linked to the gene region where the repeat is located (i.e., coding, non-coding, promotor regions) (top), without identifying the mechanisms and/or genes triggering it. DNA expansion progress to (ii) toxic RNA processes (middle) when the expanded (mutant) RNA is the main triggering cause of downstream cellular dysregulation but also to (iii) toxic protein processes, when the repeat expansion is directly affecting the translation of the final protein of the gene where the repeat is located (bottom). At this point, gain- and loss-of-function processes have been described. The *Drosophila melanogaster* organism has been modeled for several of the first NREDs identified, trying to resemble the previously mentioned mechanisms, mainly with success, but with still some challenges to target (i.e., characterizing DNA repeat instability processes, insertion of repeat expansions in the same—homolog—gene in flies). Importantly, when expanded repeats are successfully integrated and expressed and/or translated into the flies using tissue-specific genetic tools, significant phenotypic readouts have been well-characterized (right), useful for forwarding genetic- and therapeutic-based studies. *Created in Biorender*.

## Data Availability

No new data were created. This is a review manuscript with all the data provided available from published work or Pre-print servers.

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
