# Peer review of "Decoding Nucleotide Repeat Expansion Diseases: Novel Insights from Drosophila melanogaster Studies"

_ijms, 2024, doi:10.3390/ijms252111794_

Round 1

Reviewer 1 Report

Comments and Suggestions for Authors

In the review, authors want to highlight the relevance of the Drosophila model system in the field of nucleotide repeat expansion diseases (NREDs). They focus on the numerous works produced in the last seven years, providing a timely report of the research carried out both to characterize the main molecular mechanisms underlying the NREDs studied, and to find potential molecules and/or genetic modifiers capable of mitigating the symptomatology of the diseases in examination.

The review is well conceived. The model system is well introduced, the division into chapters is satisfactory, the works are correctly cited, and the scientific impact is very significant. It's a nice job. Anyway, also due to the amount of information, it's quite heavy to read, and is punctuated by several errors to correct.

Although I am not a native speaker I recommend a major revision of the English language. In my opinion several sentences need to be rewritten and overall, the whole read is quite heavy. And this is a pity considering its great scientific impact in this field, being an exhaustive and complete work.

Perhaps some tables or another image could help to lighten the reading.

I recommend publication of this manuscript after linguistic revision.

Comments on the Quality of English Language

I think the English language should be improved because it is sometimes very convoluted and non-discursive

Author Response

Thanks to the reviewer for the positive comments.

Regarding the specific suggestions for improvement:

(1) We performed a major English language revision

(2) Included Table 1, listing all the new models identified during the manuscript writing, for a better reading of the full review text.

All the changes made are highlighted as visible throughout all the text.

Hopefully, the current version is ready for publication.

Reviewer 2 Report

Comments and Suggestions for Authors

The review by Atienzar-Aroca is timely and potentially very helpful to people outside the fields, people in the field that need to be updated on the contributions of Drosophila and even Drosophilists that seek such an update.

Because i think this will be a useful review i strongly recommend the following

1. There are numerous instances where terms are not defined and that makes it vary difficult for non-afficionados to follow. for example:

line 193. What are beta breaker mutations??

line 206; what are iNeurons?

line 231: modulated ATXN1 toxicity?? modulated how?? enhance. suppress?? this is the case regarding the effects of many other interactors. For example line 251 "regulate"". which way??

line 239: what is "exp"ataxin 1??

line 326: "RNAexp and RNAtox" what are these referring to??

The list can go on and on and it pointless to continue.  Please fix these so teh reader is not obliged to look these terms up to continue reading!

3. Because the field is rather broad and varied, there an obvious need of summary tables for each category of effects Gain of function, loss of function etc.

I believe it would be best to have one summary table per category where the condition/disease is listed, the main pathologies, the fly strains used to address it, the fly phenotype(s) and genes revealed by screen to enhance/suppress the fly phenotype.

Such tables offer expedient referencing and because they are very useful they are referenced a lot... 

Comments on the Quality of English Language

The manuscript is in dire need of language editing by a native speaker. There are numerous syntax errors and odd phrases that need to be corrected for smooth reading and comprehension. However, prior to this the authors need to also correct the numerous fragmented sentences in text. They also need to go over the text carefully to correct oversights. for example:

line 54...i assume they missed a reference (REF) 

line 152: what are side effects??

Author Response

Thanks to the reviewer for the positive comments and the usefulness of the manuscript written.

Regarding the suggestions and comments provided, we have worked through all of them, making several changes, highlighted as visible editing:

(1) We have made a full-text revision of the not well-defined terms found, the suggested and the newly found going throughout all the text. 

(2) We also performed extensive work regarding language editing, done by a native English scientist (external service)

(3) We have included a summary table for the new models identified through the writing of the manuscript, considering it the most relevant data that should be easily visible and accessible throughout the whole manuscript. The table included the category of the effect, condition/disease, the name for the fly strains used, the description of them, and the references where the strains used.

Hopefully, the current version of the manuscript is ready for publication.

Round 2

Reviewer 2 Report

Comments and Suggestions for Authors

the authors did a great job in revising their manuscript. They addressed all my concerns and added the table I recommended